# Bacterial PncA improves diet-induced NAFLD in mice by enabling the transition from nicotinamide to nicotinic acid

Shengyu Feng[1], Liuling Guo[1], Hao Wang[1], Shanshan Yang[1] & Hailiang Liu [1,2✉]

Nicotinamide adenine dinucleotide ($NAD^+$) is crucial for energy metabolism, oxidative stress, DNA damage repair, longevity regulation, and several signaling processes. To date, several $NAD^+$ synthesis pathways have been found in microbiota and mammals, but the potential relationship between gut microbiota and their hosts in regulating $NAD^+$ homeostasis remains largely unknown. Here, we showed that an analog of the first-line tuberculosis drug pyrazinamide, which is converted by nicotinamidase/pyrazinamidase (PncA) to its active form, affected $NAD^+$ level in the intestines and liver of mice and disrupted the homeostasis of gut microbiota. Furthermore, by overexpressing modified PncA of *Escherichia coli*, $NAD^+$ levels in mouse liver were significantly increased, and diet-induced non-alcoholic fatty liver disease (NAFLD) was ameliorated in mice. Overall, the PncA gene in microbiota plays an important role in regulating $NAD^+$ synthesis in the host, thereby providing a potential target for modulating host $NAD^+$ level.

---

[1] Institute for Regenerative Medicine, Shanghai East Hospital, Tongji University School of Medicine, 200123 Shanghai, China. [2] Key Laboratory of Xinjiang Phytomedicine Resource and Utilization of Ministry of Education, College of Life Sciences, Shihezi University, 832003 Shihezi, China. ✉email: hailiang_1111@ tongji.edu.cn

As early as 110 years ago, NAD+ was discovered by the British biochemists Arthur Harden and William John Young as a coenzyme involved in yeast-mediated alcohol fermentation[1]. Over the next ~30 years, the chemical composition of this coenzyme was determined, and it was found to participate in several redox reactions together with NADH. Although the role of NAD+ in oxidation–reduction reactions is well understood, it was not until the last 10 years that the function of NAD+ was completely elucidated. The sirtuin (SIRT) family, poly (ADP-ribose) polymerases (PARPs), and cyclic ADP-ribose synthases (cADPRSs) are NAD+-dependent enzymes; so NAD+ regulates downstream metabolic pathways by influencing the activity of these enzymes and acting as a metabolic sensor in cells[2–5]. In addition, these NAD+-dependent proteins play an important role in various biological processes, such as metabolism, signal transduction, oxidative stress, cognitive decline, and other aging-related physiological processes. In recent years, important studies have revealed that NAD+ binds to the 5′ end of mRNA, thereby regulating the transcription initiation of genes. However, there is still no definitive conclusion on how this process is regulated or its significance for organisms[6,7].

NAD+ is a small molecule required by almost all organisms and is one of the most abundant molecules in the human body, participating in more than 500 different enzymatic reactions[8]. In mammals, there are three main synthesis pathways for NAD+: the de novo synthesis pathway using tryptophan, the salvage pathway using nicotinamide (NAM), and the Preiss–Handler pathway using nicotinic acid (NA). Because there are several mechanisms to synthesize NAD+ in mammals, the specific pathways used for different tissues and the most effective pathway remain unclear[9,10]. In the intestinal flora, there is an additional important pathway that converts NAM into NA, combining the salvage and Preiss–Handler pathways, which is known as deamidation and is catalyzed by PncA[11]. This process has recently been shown to be an important step for the intestinal flora to regulate host NAD+ level[12]. This pathway catalyzed by PncA is considered to have been abandoned during biological evolution because no homolog of this gene has been identified in higher mammals. As a result, no enzyme catalyzing the conversion of NAM to NA has been found in mammals to date[9]. Based on these characteristics, pyrazinamide, a drug that requires PncA to convert into its active form, was developed to treat Mycobacterium tuberculosis[13].

NAFLD is the most common chronic liver disease and is closely related to metabolic syndrome. With substantial changes in diet and lifestyle, the prevalence of NAFLD has increased significantly worldwide[14]. However, there are currently no effective FDA-approved drugs available for clinical use. Therefore, exploring promising therapeutic targets or strategies remains a priority[15]. Several studies have shown that supplementation with the NAD+ precursors nicotinamide ribonucleotide (NR), nicotinamide mononucleotide (NMN), and ACMSD inhibition (an enzyme that blocks the de novo NAD+ synthesis pathway) significantly improve NAFLD in mice[16–18], suggesting NAD+ regulation in the diseased liver as a potential therapeutic target. During the development of NAFLD, the intestinal flora undergoes significant changes[19], and PncA in the intestinal flora plays an important role in NAD+ synthesis in mouse liver. Therefore, this study aimed to explore the significance of PncA in regulating mouse NAD+ level and its potential application value in improving NAFLD in mice.

Our research indicated that pyrazinecarbonitrile (PCN) inhibited PncA activity and affected NAD+ levels in the intestines and liver of mice, and disrupted the homeostasis of gut microbiota. Overexpression of modified PncA in mice significantly increased NAD+ level in the liver and improved diet-induced NAFLD.

## Results

**PncA inhibitor PCN disrupts gut microbiome homeostasis and reduces host NAD+ level.** We summarized the NAD+ synthesis and consumption pathways in mammals and human intestinal flora (Fig. 1). We could see that PncA is absent in mammals but is highly conserved in different bacterial phyla (Supplementary Fig. S1), indicating its importance in the bacterial synthesis of NAD+. Supplementary Fig. S1 also shows other genes related to NAD+ synthesis in different types of bacteria.

To verify whether the PncA in microbiota affects NAD+ synthesis in the host, we used the PncA inhibitor PCN, which is an analog of the commonly used antibiotic pyrazinamide to treat tuberculosis. PCN has been reported to have a strong inhibitory effect on the PncA enzyme activity of tuberculosis[20]. The protein structure alignment showed the active catalytic sites of PncA are conserved in bacteria (Supplementary Fig. S2a), so we speculated that PCN would also function in other bacteria. An enzyme assay of purified PncA of E. coli also showed that PCN strongly inhibited its deamination activity (Supplementary Fig. S2b and c). To verify the influence of PCN on bacterial growth in vitro, we selected several bacteria with different NAD+ synthesis pathways, including Bifidobacterium longum, which has both de novo synthesis and Preiss–Handler pathways; Akkermansia muciniphila, which only has the de novo synthesis pathway; and Lactobacillus salivarius and Streptococcus gordonii, which only have the Preiss–Handler pathway (Supplementary Fig. S1). These three types of bacteria are common gut flora in mammals. The growth rates of A. muciniphila and B. longum were almost unaffected after PCN treatment (Fig. 2a), while the growth of L. salivarius and S. gordonii, which depended only on the Preiss–Handler pathway, was strongly inhibited. This demonstrates that PCN has a strong inhibitory effect on bacteria that depend on the Preiss–Handler pathway. Although B. longum has the PncA gene, PCN had a minimal effect on its growth, indicating that B. longum mainly synthesizes NAD+ through the de novo pathway.

In the in vivo experiment, we treated mice with PCN and collected the feces of each group on the last day of the experiment. We then used high-throughput sequencing technology to analyze the changes in the intestinal flora of mice. Principal component analysis (PCA) indicated that PCN treatment had a significant impact on the gut microbes in mice (Fig. 2b). Surprisingly, mice treated with PCN exhibited a significant increase in bacterial richness compared with control mice (Fig. 2c). As a potential explanation, we speculate that PCN disrupts the original balance in the intestinal flora, resulting in the massive expansion of some bacteria. At the species level, PCN significantly increased the abundance of Helicobacter hepaticus, Clostridium cocleatum, and B. pseudolongum (Supplementary Fig. S3a). Besides, the abundance of Bacteroidales increased significantly, whereas the abundance of Clostridiales decreased significantly (Supplementary Fig. S3b). However, we did not find any relationship between those bacteria and their NAD+ synthesis pathway, owing to the complex composition and interdependence of gut microbiota. Moreover, PCN significantly inhibited electron transfer, respiration, and other pathways closely related to NAD+ in the intestinal flora (Supplementary Fig. S3c). Supplementary Fig. S3d shows that NAM increased while NA decreased in the feces of mice after PCN treatment. Together, these findings show that PCN has a significant impact on intestinal flora.

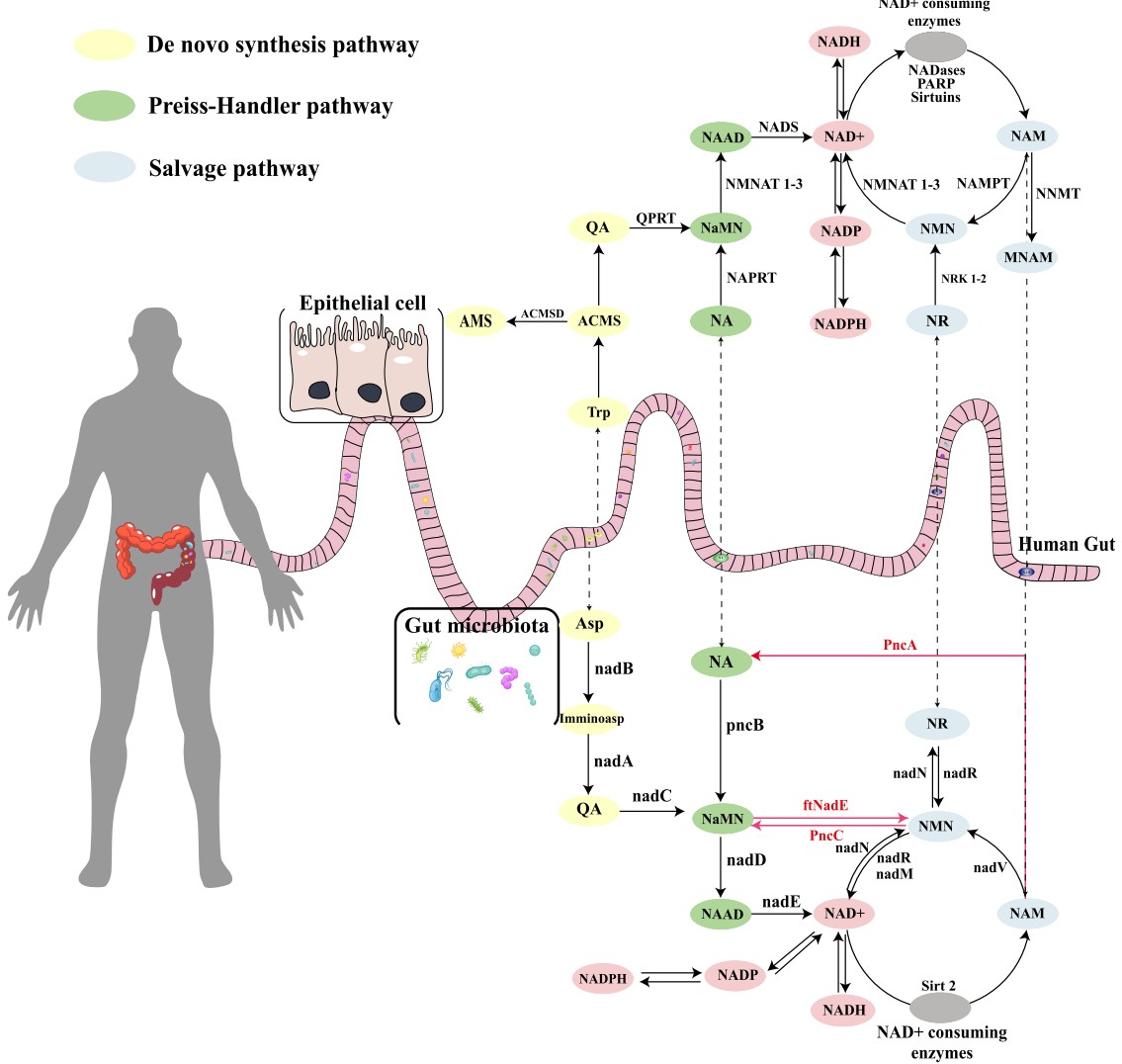

**Fig. 1 The synthesis and consumption pathways of NAD⁺ in mammals and intestinal flora.** Two main pathways involved in mammalian NAD synthesis: de novo and salvage pathway. The former pathway converts Trp to QA through the kynurenine pathway and synthesizes NAD⁺. The main precursors of NAD⁺ are NAM, NA, NR, and NMN. NAD⁺-consuming enzymes mainly include SIRTs, PARPs, and CD38, and the metabolites after they use up NAD all contain NAM. In the intestinal flora, the de novo synthesis of NAD⁺ mainly depends on Asp rather than Trp. Although different in the de novo pathway, they share the same precursors for NAD⁺ synthesis. Dependence on the same precursors must lead to competition or cooperation in NAD⁺ metabolism between mammals and their microbiota. The synthetic pathway marked by the red line is specific to gut flora. Trp tryptophan, ACMS α-amino-β-carboxymuconate-ε-semialdehyde, QA quinolinic acid, AMS α-amino-β-muconate-ε-semialdehyde, NaAD nicotinic acid adenine dinucleotide, MNAM methylation of NAM, Asp aspartic acid, ImminoAsp immino-aspartate, QPRT nicotinate-nucleotide pyrophosphorylase, NMNAT Nicotinamide/nicotinic acid mononucleotide adenylyltransferase, NADS glutamine-dependent NAD⁺ synthetase, NNMT nicotinamide N-methyltransferase, NRK nicotinamide riboside kinase, NADP nicotinamide adenine dinucleotide phosphate. Figure 1 was created by Adobe Illustrator.

Because *PncA* in the intestinal flora plays an important role in the synthesis of NAD⁺ in the host intestine and liver, we explored the effects of PCN treatment on the level of NAD⁺ in the liver and intestine of host mice. PCN reduced the utilization efficiency of NAM in the liver and intestine of the host, resulting in lower NAD⁺ level (Fig. 2d and e). Given that PncA catalyzes the conversion of NAM to NA, we speculated that NAD⁺ synthesis using NA is more efficient than that using NAM in the liver and intestine. It is well known that NAM is produced when NAD⁺-dependent enzymes consume NAD⁺, and additional supplementation does not enhance NAD⁺ synthesis. Surprisingly, in the case of NAM supplementation, PCN also reduced the level of NAD⁺ in the hippocampus compared with the PBS group (Fig. 2f). According to RNA sequencing (RNA-seq) results, PCN affected the expression of numerous genes in the liver

(Supplementary Fig. S4a) and significantly affected host immune processes, such as the intestinal immune network for IgA production and antigen processing (Supplementary Fig. S4b). To demonstrate that PCN regulates NAD⁺ level in the host through its effect on the intestinal flora rather than directly affecting the host, 293T, HepG2 cells, and antibiotic-treated mice were treated with PCN. The results showed that PCN had no effect on the growth and NAD⁺ level of human cells as well as antibiotic-treated mouse colon and liver (Supplementary Fig. S5a–d).

**Escherichia coli overexpressing *PncA* affects host liver NAD metabolism.** Our previous experiments and the research of Shats et al. showed that the *PncA* gene in the intestinal flora plays an important role in regulating host NAD⁺. Therefore, we attempted

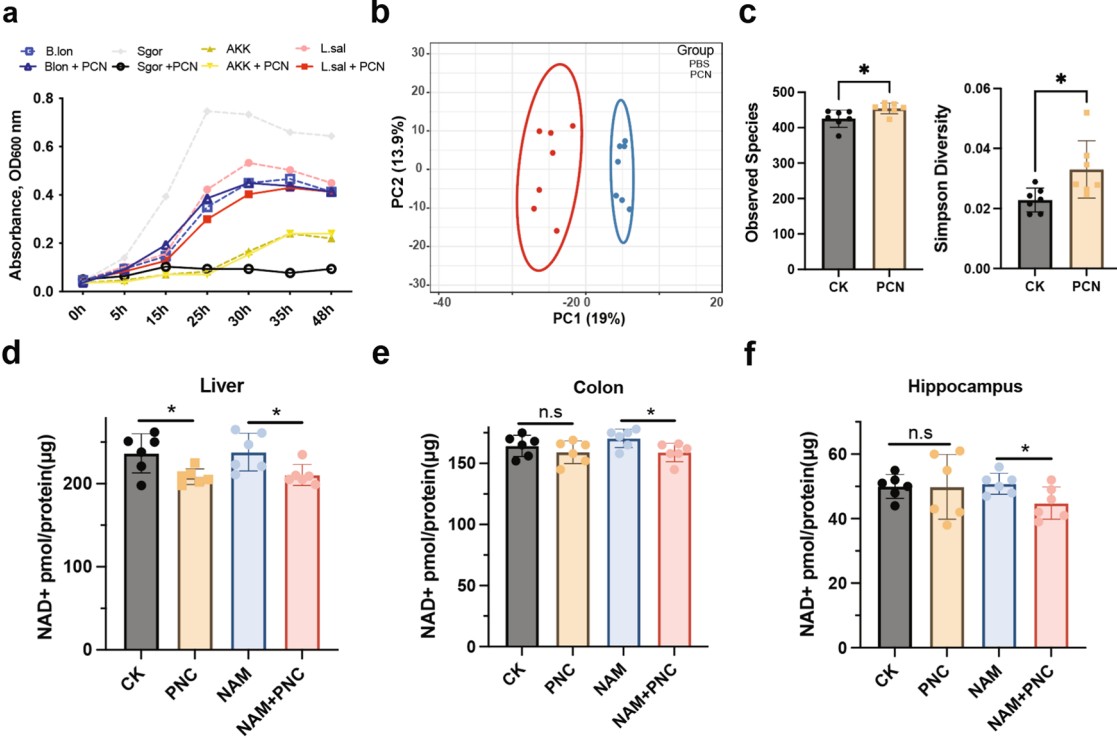

**Fig. 2 PncA inhibitor disrupts intestinal microbiome homeostasis and reduces host NAD$^+$ level. a** Effects of PCN on the growth of bacteria with different NAD$^+$ synthesis pathways. **b** PCA diagram of the intestinal flora in mice treated with PCN ($n = 7$). **c** Intestinal microbial species and diversity after PCN treatment in mice ($n = 7$, $*p < 0.05$). **d–f** NAD$^+$ level in the liver, intestine, and hippocampus of mice ($n = 6$, $*p < 0.05$, n.s = no significance).

to promote the synthesis of mammalian NAD$^+$ using a new approach. We first wanted to supplement bacteria that specifically depend on the deamidation NAD$^+$ synthesis pathway. However, controlling univariate variables for this experiment was challenging, so we constructed *E. coli* with overexpression of PncA (PncA-OE) and used normal *E. coli* strains with only vector (PncA-WT) as the control. Based on the gene expression, protein quantification, and enzyme assay of PncA in different *E. coli* (Supplementary Fig. S6a–c), *PncA* was functionally induced in the PncA-OE group.

First, we performed an in vitro experiment. NAM (0.5 mM) was added to the culture medium of *E. coli*, which was cultured until optical density (OD) reached 1.0. Then bacteria were centrifuged, and the supernatant was collected for metabolome analysis (Fig. 3a). As expected, PncA-OE *E. coli* released more NA in the bacterial culture medium than PncA-WT *E. coli* did (Fig. 3b). To facilitate the colonization of exogenous *E. coli* in the intestines of mice in vivo, we first treated mice with a cocktail of antibiotics for 5 days to reduce endogenous bacteria. Next, mice were gavaged with PncA-OE and PncA-WT *E. coli* and divided into NAM-supplemented and non-NAM-supplemented groups (Fig. 3c). Previous studies showed that the efficiency of NA utilization for NAD$^+$ synthesis in the liver is higher than that of NAM[21]. We found that the NAD$^+$ level in mouse liver in the PncA-OE *E. coli* group was higher than the control in the condition of NAM supplement (Fig. 3d).

Enhancing the synthesis of NAD$^+$ in liver significantly improves NAFLD. Therefore, we constructed NAFLD model mice using a methionine- and choline-deficient (MCD) diet to verify whether PncA-OE *E. coli* could ameliorate NAFLD. MCD diet rapidly decreased the body weight of the mice (Fig. 4a) and caused liver physiological lesions (Fig. 4b). MCD diet also significantly reduced liver NAD$^+$ and ATP levels (Fig. 4c and d) and increased triglycerides in the liver (Fig. 4e). In addition, after

supplementation with PncA-OE *E. coli*, NAD$^+$ and ATP level in the liver were slightly increased compared with the control group but not significantly (Fig. 4c and d). PncA-OE *E. coli* did not resolve the lesions in the liver (Fig. 4b). That might have been caused by the low colonization rate of *E. coli* in mice and low transformation efficiency of NAM to NA by PncA-OE *E.coli* because we did not observe accumulation of NA in feces during supplementation with PncA-OE *E. coli* (Supplementary Fig. S6d).

**PncA overexpression in the liver by adeno-associated virus improves hepatic lesions in mice.** We found that supplementation with PncA-OE *E. coli* was not effective in relieving NAFLD. However, we speculate that this may have been because of the limited colonization ability of bacteria to promote liver NAD$^+$. Therefore, instead of using bacteria, we optimized the sequence of *PncA* of *E. coli* for direct expression in mammals. A liver-specific adeno-associated virus (AAV) was constructed to carry *PncA* into mice via tail vein injection. PncA was highly expressed in mouse liver (Fig. 5a). Intriguingly, the NAD$^+$ level in the AAV-PncA group was significantly higher than that in the vector group and even approximately five times higher than that in the normal diet group (Fig. 5b). This was particularly surprising because the widely used and highly efficient NAD$^+$ precursors NR and NMN only increased the level of NAD$^+$ in the liver by 1.5–2 fold[10]. However, previous studies have not shown that NA had such a strong NAD$^+$ booster effect on mouse liver[22,23]. By comparing our results with others, we speculate that the effect of NA on NAD$^+$ synthesis may be limited by the absorption efficiency of NA by liver cells. *PncA* expression in the liver promotes the transformation of NAM to NA in cells, maximizing the synthesis of NA to NAD$^+$. This suggests that the nicotinamidase PncA, which appears to have been abandoned during the evolution of the NAD$^+$ synthesis pathway, has the potential for NAD$^+$ synthesis in mammals.

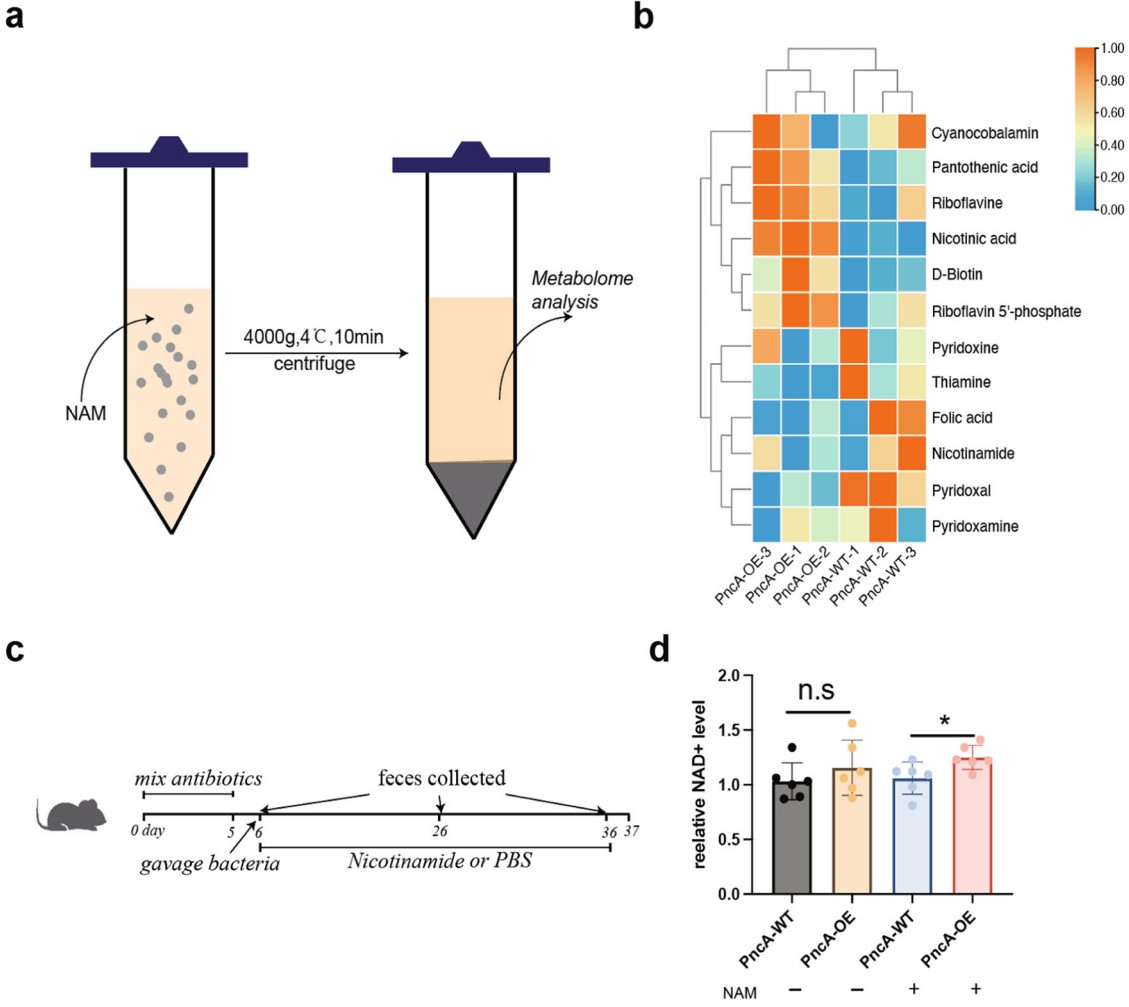

**Fig. 3 PncA-overexpressing *E. coli* affect host liver NAD metabolism. a** Schematic diagram of the in vitro experiment (created by Adobe Illustrator).
**b** Heatmap of the metabolites in the bacterial culture medium. **c** Schematic diagram of the mouse experiment. **d** Liver NAD$^+$ level after colonizing mice with different genotypes of *E. coli* ($n = 6$, $^*p < 0.05$, n.s = no significance).

In addition, compared with the vector group, mice expressing PncA showed a significant increase in ATP level and a decreased triglyceride content after supplementation with the MCD diet (Fig. 5c and d). Furthermore, the results of Oil Red O and H&E staining showed that overexpression of *PncA* in mouse liver improved NAFLD-induced pathological changes (Fig. 6a). RNA-seq analysis showed that the expression of some genes involved in lipid metabolism was significantly changed (Supplementary Fig. S7a), and gene set enrichment analysis revealed that PncA increased the expression of genes mainly involved in the PPAR signaling pathway, fatty acid degradation, and other pathways that promote fat metabolism (Fig. 6b). Based on the metabolome analysis, PCA revealed significant differences between the PncA and vector groups (Supplementary Fig. S7b), and the volcano diagram demonstrated a large number of differential metabolites between the two groups (Supplementary Fig. S7c). The NA concentration in the liver of the PncA group was significantly increased (Fig. 6c), and the levels of other small molecules involved in nicotinate and nicotinamide metabolism were also significantly altered (Supplementary Fig. S7d). Metabolites that differed between the two groups were mainly enriched in the nicotinate and nicotinamide metabolic pathways, followed by phenylalanine, tyrosine, and tryptophan biosynthesis and other pathways related to lipid metabolism (Supplementary Fig. S7e). The raw metabolome analysis data are shown in Supplementary

Data 1. To our surprise, the highly expressed genes in the PncA group were also significantly enriched in the Th1 and Th2 cell differentiation pathways, indicating that *PncA* genes may play an important role in T cell differentiation (Fig. 6d).

To verify that direct NA supplementation in mice did not recapitulate the effect of expressing PncA in the liver, we treated mice with the MCD diet and NA. The body weight of mice decreased throughout the experiment (Supplementary Fig. S8a), and NA had no obvious effect on the NAD$^+$ level in the liver (Supplementary Fig. S8b). Therefore, direct NA supplementation does not increase the level of mouse NAD$^+$.

## Discussion
Coenzyme NAD$^+$ plays a key role in cellular biology and adaptive stress responses. Its depletion is a basic feature of aging and may lead to various chronic diseases[24]. NAD$^+$ supplementation alleviates several aging-related diseases and even prolongs the life-span of mice[25]. Maintaining NAD$^+$ level is essential for the function of high-energy-demanding cells and mature neurons[26]. NAD$^+$ levels are substantially decreased in major neurodegenerative diseases, such as Alzheimer's disease, Parkinson's disease, and muscle atrophy[27]. Increasing evidence has shown that NAD$^+$ is significantly reduced in various tissues during aging, and determining how to efficiently increase cellular NAD$^+$ levels through physiological and pharmacological methods and prevent

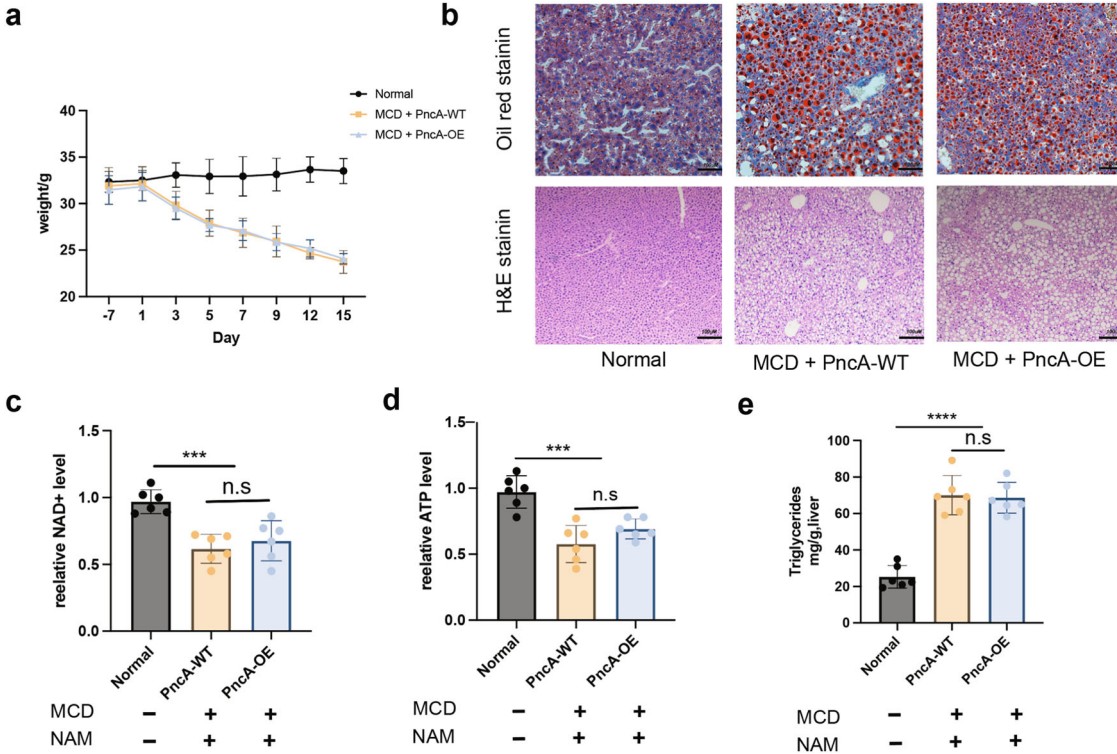

**Fig. 4 PncA-overexpressing *E. coli* did not significantly ameliorate mouse NAFLD. a** Body weight of MCD-induced NAFLD model mice ($n = 6$). **b** Representative Oil Red O staining (top) and hematoxylin and eosin (H&E) staining (bottom) of mouse liver sections. **c** Liver NAD$^+$ level in NAFLD model mice treated with different bacteria. **d** Relative content of ATP in mouse liver. **e** Content of triglyceride in mouse liver. ($n = 6$, ***$p < 0.001$, ****$p < 0.0001$, n.s = no significance). Scale bars 100 μm.

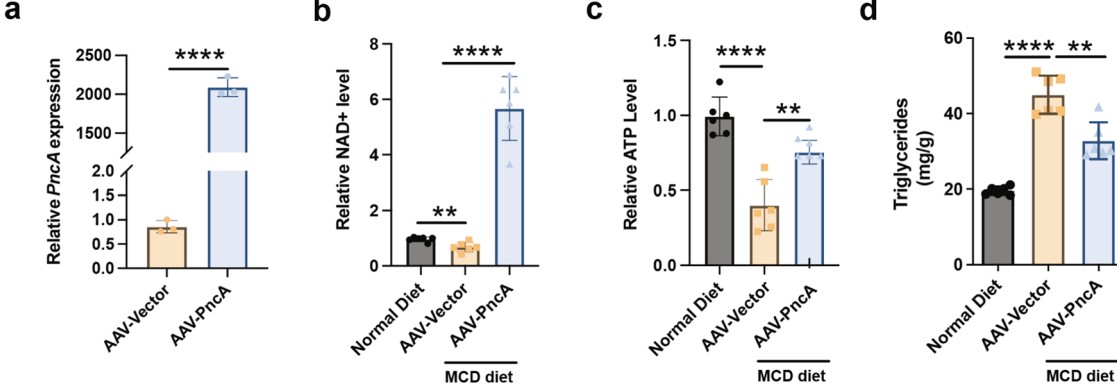

**Fig. 5 Improvement of NAD$^+$ in mouse liver by PncA overexpression via liver-specific AAV. a** PncA expression in mouse liver in the control and PncA groups ($n = 3$, ****$p < 0.0001$). **b** Relative content of NAD$^+$ in mouse liver. **c** Relative content of ATP in mouse liver. **d** Content of triglyceride in mouse liver. ($n = 6$, **$p < 0.01$, ****$p < 0.0001$).

age-related diseases has become a hot research topic in recent years. Similarly, identifying new and efficient methods to promote NAD$^+$ synthesis is also vital.

Currently, the most efficient NAD$^+$ synthesis pathway in mammals is a controversial issue. In addition, different organs depend on different NAD$^+$ precursors[23]. Studies have reported that the liver and kidneys use all three NAD$^+$ synthesis pathways; the spleen, small intestine, and pancreas mainly rely on the salvage of NA and NAM, while the heart, lungs, brain, muscles, and white adipose tissue primarily use the NAM pathway[28]. Regarding the most efficient precursor, some articles have reported that the efficiency of NAD$^+$ synthesis with NA is higher than that with NAM, but there is still no clear conclusion[29]. However, the most widely used NAD$^+$ precursors are NR and NMN. After

these two precursors enter the cell, NAD$^+$ is synthesized through one or two enzymatic reactions, and it avoids being affected by rate-limiting enzymes, such as nicotinate phosphoribosyltransferase (NAPRT) and nicotinamide phosphoribosyltransferase (NAMPT). These two precursors have been used for various interventions, such as alleviating neurodegenerative diseases, improving hearing, treating diabetes and NAFLD, delaying aging, and extending lifespan[30–34].

Cells have different absorption efficiencies for various NAD$^+$ precursors. NAM directly enters the cell through free diffusion, whereas NA, NR, NMN, and other precursors require the assistance of membrane proteins, resulting in a lower absorption efficiency compared with NAM[35]. However, NAM is believed to be present in sufficient amounts in the body because NAD$^+$ is

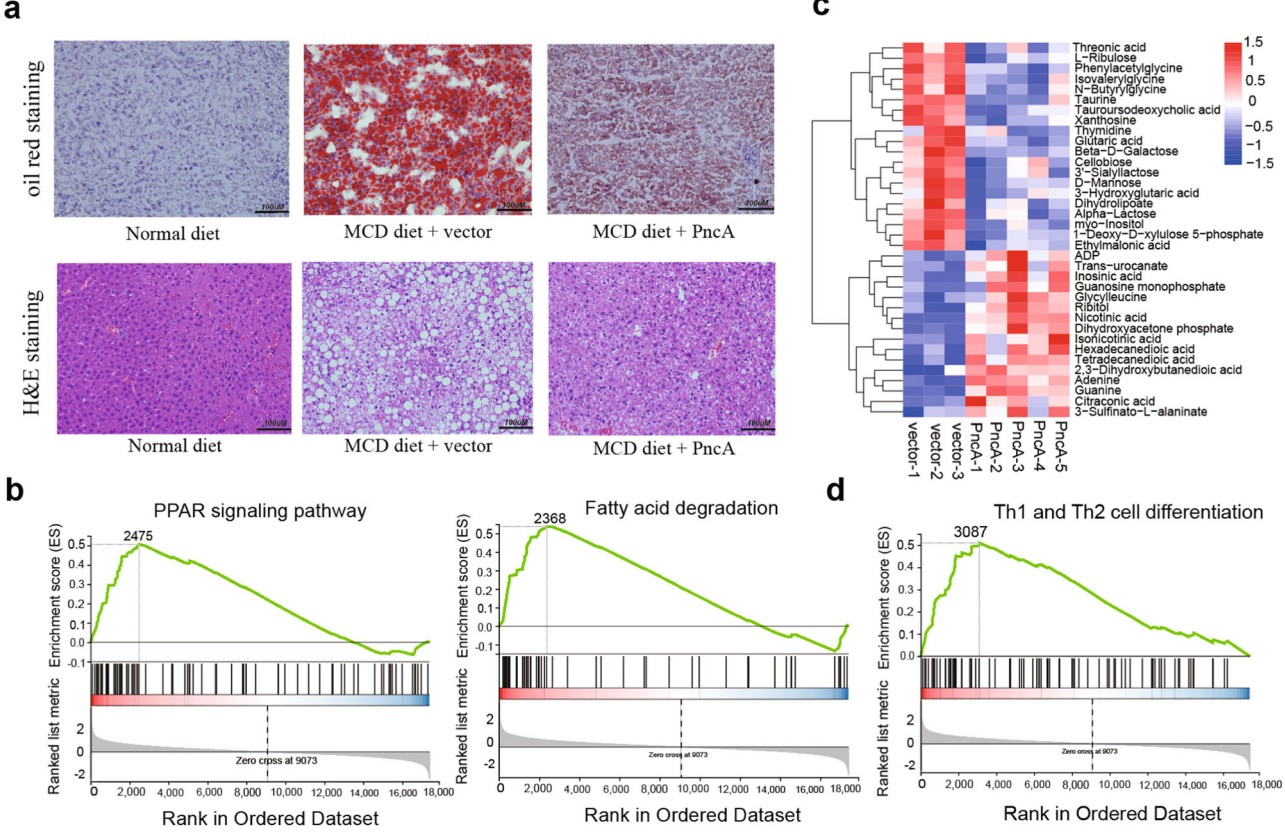

**Fig. 6 Improvement in mouse liver lesions by PncA overexpression via AAV. a** Representative Oil Red O staining (top) and H&E staining (bottom) of mouse liver sections. **b** Gene set enrichment analysis (GSEA) of RNA-seq results. **c** Heatmaps of metabolites with significant differences between the PncA and vector groups. **d** GSEA of RNA-seq results. Scale bars 100 μm.

decomposed to produce NAM, which re-enters the NAD$^+$ synthesis pathway. In addition, NAM is an inhibitor of the SIRTs family. Excessive concentrations of NAM inhibit the activity of SIRT2 and shorten the lifespan of *Saccharomyces cerevisiae*[36]. However, PncA overexpression in *Drosophila* protects neurons and extends their lifespan[37]. *PncA* also increases the lifespan of *Caenorhabditis elegans*[38]. In this study, we used liver-specific AAV to express PncA in the liver, an organ that relies on multiple NAD$^+$ precursors. In this way, we replenished the liver with the nicotinamidase that was discarded during evolution. The results showed that the Preiss–Handler pathway was highly active in mammalian cells, and this approach increased the level of NAD$^+$ to a greater extent than NR or NMN, which indicated the high efficiency of NA for NAD$^+$ synthesis.

Furthermore, the efficiency of using NA for NAD$^+$ synthesis appears to be highest in the liver. In addition, PncA can process excessive NAM in the liver, thereby relieving the inhibition of SIRT1 by NAM and increasing the activity of SIRT1, which needs further verification. Based on the above results, we speculate that *PncA* also substantially increases NAD$^+$ levels in other organs that rely on NA for NAD$^+$ synthesis. Moreover, we demonstrated that *PncA* significantly improved NAFLD in mice, indicating that *PncA* provides a promising potential target for treating various diseases related to NAD$^+$ deficiency. There are some limitations in our work; germ-free mice would be a more ideal research model for bacteria supplementary experiment, and a constitutive bacterial expression vector would be a high-efficiency method for protein overexpression in vivo which may explain why we did not observe the increase of NA in feces. Besides, further research should be done to evaluate the potential side effects caused by excessive elevation of NAD$^+$ after *PncA* overexpression.

## Methods

**Materials**. NAM (HY-B0150), NA (HY-B0143), and NR (HY-123033) were obtained from MedChemExpress (Monmouth Junction, NJ, USA). PCN (243-369-5) was obtained from Sigma-Aldrich (St. Louis, MO, USA), and the MCD diet (TD.90262) was obtained from Harlan Teklad. Brain heart infusion (BHI) (HB8297-5), MRS (HB0384-1), and TPY (HB8570) were obtained from Hopebio-Technology (Qingdao, China).

**Cell lines**. 293T female embryonic kidney cells and HepG2 liver cancer cells were obtained from ATCC and cultured in Dulbecco's modified Eagle's medium (Thermo Fisher Scientific, Waltham, MA, USA) supplemented with 10% fetal bovine serum (FBS) and 100 mg/ml penicillin/streptomycin. Each cell line was maintained in a 5% CO$_2$ atmosphere at 37 °C. The mycoplasma contamination status of all cultures was monitored monthly by PCR.

**Bacterial strains**. *Akkermansia muciniphila* (ATCC BAA-835) and *L. salivarius* (ATCC 1174) were cultured anaerobically in BHI and MRS medium, respectively. *Bifidobacterium longum* (1.2186) and *S. gordonii* (1.2496) were obtained from the China General Microbiological Culture Collection Center (Beijing, China) and cultured anaerobically in TPY and BHI medium, respectively.

**Mice**. Wild-type C57BL/6J mice were purchased from Cyagen Biosciences Inc (Suzhou, China). Mice were maintained under a 12-h light/dark cycle and fed a standard chow diet in the specific pathogen-free facility at the Laboratory Animal Research Center, Tongji University. For bacterial and AAV infection, 8- to 10-week-old female C57BL/6J mice were used. All experiments were carried out following the national guidelines for the housing and care of laboratory animals (Ministry of Health, China), and the protocol complied with institutional regulations after review and approval by the Institutional Animal Ethics Committee at the Laboratory Animal Research Center, Tongji University (TJAA09220102). Mice were acclimatized for a period of 7 days before the initiation of the experiment.

**Mouse infection**. Eight-week-old C57BL/6 female mice were fed with either regular water or autoclaved water containing an antibiotic cocktail (1 g/L ampicillin, 1 g/L neomycin, 1 g/L metronidazole, and 500 mg/L vancomycin) for 5 days and then given regular water. *Escherichia coli* strains were cultured as described above

and quantified prior to infection. For bacterial colonization assays, the mice were infected intragastrically with $1 \times 10^9$ cfu PncA-OE or PncA-WT E. coli strains (in 0.2 ml PBS) every 3 days until the end of the experiment. For the group with $NAD^+$ precursors, NAM (400 mg/kg/day) and NA 400 mg/kg/day were delivered via gavage. AAV ($1 \times 10^{12}$ pfu/ml in 100 μl PBS) expressing *PncA* (AAV-PncA) or vector (AAV-vector) were injected into mice via their tail vein. Animals were sacrificed 60 days after injection. The liver was removed and stored at −80 °C until use.

**MCD diet-induced NAFLD.** Eight-week-old C57BL/6J female mice were used. After acclimatization, the mice were randomized based on their body weight and separated into three groups; two receiving the MCD diet and one receiving a normal control diet. For the bacterial experiment, the MCD diet was supplemented 30 days after the first bacterial colonization. For the AAV experiment, the MCD diet was supplemented 40 days after the AAV injection. Then, 2.5 weeks after the beginning of the special diets, animals fasted for 4 h and were anesthetized with isoflurane. Isolated tissues were snap-frozen in liquid nitrogen and stored at −80 °C for later experiments. The experiment was performed twice.

**Purification of PncA and enzyme activity assay.** His-tag Protein Purification Kit (P2226) from Beyotime (Beijing, China) was used to purify PncA. PncA activity was determined by the ammonia production detected by the ammonia assay kit (MAK310) from Sigma-Aldrich. In a typical assay, 100 mM HEPES, pH 7.4, containing 500 μM nicotinamide and 37.5 nM PncA were combined at 27 °C. The addition of enzyme initiated the reaction, and ammonia was detected after 15 min. The amount of conversion of NAM by PncA was calculated by ammonia production. As for the bacterial enzyme assay, lysis of E. coli was used.

**PCN experiment.** Eight-week-old C57BL/6 female mice were gavaged with either PBS or PCN (150 mg/kg) every day for 2 weeks. Fresh feces were collected rapidly on day 15 and stored at −80 °C until processing for DNA extraction and other processes. Animals were killed at the end of the experiment. Tissues were removed and stored at −80 °C until use. PCN concentration of 100 μg/ml was used for the cell and bacterial experiments.

**Construction of PncA-OE E. coli.** For PncA-OE E. coli, the PCR product of a His-tagged PncA was cloned into the pET-28a vector to construct the expression plasmid, which was validated by Sanger sequencing. The plasmid pET-28a-PncA was transformed into E. coli BL21 (DE3), which was cultured in LB broth with kanamycin (50 μg/ml) at 37 °C and shaking at 220 rpm. Isopropyl β-D-1-thiogalactopyranoside (0.5 mM) was added to the LB broth to induce PncA expression during the logarithmic growth phase (OD600 ~0.6). The bacteria were harvested when OD600 was >1. Cells were pelleted by centrifuging for 10 min at 5000×g and resuspended in PBS. The bacteria were counted and stored in 30% glycerol at −80 °C. The overexpression of PncA in E. coli was validated by qPCR and western blotting.

**Cell transfection.** Transient transfection was performed using LipoFiter 3.0 (Hanbio Biotechnology Co. Ltd, Shanghai, China).

**Preparation of AAV.** 293T cells were used to package adenoviruses using a three-plasmid system, including pAAV-RC, pHelper, and shuttle plasmids (with or without the target gene). 293T cells were subcultured into 100-mm plates for transfection. Transfection was performed when the cell density reached 80–90% with the following transfection complex reagents: pAAV-RC 10 μg, pHelper 20 μg, shuttle plasmid 10 μg, and Lipofiter™ (HB-TRCF-1000, Hanbio Biotechnology Co. Ltd, Shanghai, China) 120 μl. Fresh complete medium containing 10% FBS was replaced 6 h after transfection. Seventy-two hours after transfection, cells containing AAV particles were gently removed using a cell scraper, collected into a 15-ml centrifuge tube, and centrifuged at 150×g for 3 min. Cells were collected, and the culture supernatant was removed. Cells were washed with PBS once and resuspended with 300 μl PBS. The cells were frozen and thawed in liquid nitrogen and 37 °C three times and centrifuged at 2000×g for 5 min at 4 °C to remove cell debris. The lysis supernatant containing AAV particles was collected, and 0.1 μl Benonase (9025-65-4, Merck, Darmstadt, Germany) was added to each 1 ml of crude virus extract to remove the cell genome and plasmid DNA. The cells were centrifuged at 600×g for 10 min at 4 °C, and the supernatant was collected for column purification (V1469-01, Biomiga, San Diego, CA, USA). The 4-ml AAV liquid samples purified by the column were added to the ultrafiltration tube and centrifuged at 1400×g for 30 min to obtain approximately 1 ml of purified AAV, which was stored at −80 °C until use.

**Identification of homologs of $NAD^+$-related genes in different bacteria.** We selected some common mammalian microbial flora and pathogenic bacteria from different classifications and explored the genes associated with $NAD^+$ synthesis in their genomes. Quinolinate synthetase (nadA) catalyzes the second step of the de novo biosynthetic pathway of pyridine nucleotide formation, which contains a nadA domain. A hidden Markov model (HMM) of nadA domain was downloaded from pfam[39] and used to identify nadA homologous proteins using the hmmsearch function from the HMMer 3.1 package[40] against the bacterial proteome that we selected. Matches with E ≤ $10^{-5}$ and annotation including quinolinate synthetase were recognized as candidates of nadA. BlastP (https://blast.ncbi.nlm.nih.gov/Blast.cgi?PAGE=Proteins) was used to screen the candidates using the threshold of E ≤ $10^{-5}$, and the proteins that passed these two thresholds were recognized as nadA. The identification of nadC, nadD, nadE, PncA, PncB, and Sirt2 was similar to that of nadA. However, nadB, nadR, and nadN had no signature domain, so the HMM method was not appropriate for identifying those genes. Protein sequences of those genes of E. coli were used as seeds submitted to BlastP against the bacteria proteome we selected; proteins with an E value less than $10^{-5}$ and appropriate annotation were recognized as corresponding genes.

**Untargeted and targeted vitamin B metabolomics and LC-MS/MS.** For bacterial supernatants, the samples were thawed in an ice-water bath and vortexed for 30 s. A 100-μl aliquot of each individual sample was transferred to an Eppendorf tube. After adding 400 μl extract solvent (precooled at −40 °C, acetonitrile–water, 5:3, containing 0.125% formic acid and isotopically labeled internal standard), the samples were vortexed for 30 s. The samples were sonicated for 15 min in an ice-water bath, followed by incubation at −20 °C for 1 h and centrifugation at 12,000 rpm (Relative centrifugal force (RCF) = 13,800×g) and 4 °C for 15 min. A 400-μl aliquot of the supernatant was evaporated to dryness under a gentle stream of nitrogen and reconstituted in 80 μl 1% methanol/water (v/v). After the samples were centrifuged at 12,000 rpm (RCF = 13,800×g, R = 8.6 cm) for 15 min at 4 °C, the clear supernatant was subjected to LC-MS/MS analysis.

For mouse feces, an aliquot of each individual sample was weighed and transferred to an Eppendorf tube. After adding two small steel balls and 1000 μl extraction solution (precooled at −40 °C, 50% acetonitrile containing 0.1% formic acid and isotopically labeled internal standard), the samples were vortexed for 30 s, homogenized at 40 Hz for 4 min, and sonicated for 5 min in an ice-water bath. The homogenization and sonication cycles were repeated three times, followed by incubation at −20 °C for 1 h and centrifugation at 12,000 rpm (RCF = 13,800×g, R = 8.6 cm) for 15 min at 4 °C. An 800-μl aliquot of the supernatant was evaporated to dryness under a gentle stream of nitrogen and reconstituted in 80 μl 1% methanol/water (v/v). After the samples were centrifuged at 12,000 rpm (RCF = 13,800×g, R = 8.6 cm) for 15 min at 4 °C, the clear supernatant was subjected to LC-MS/MS analysis.

For mouse liver, 50-mg samples were weighed and placed in an Eppendorf tube, and 1000 μl extract solution (methanol: acetonitrile: water = 2:2:1, with an isotopically labeled internal standard mixture) was added. The samples were homogenized at 35 Hz for 4 min and sonicated for 5 min in an ice-water bath. The homogenization and sonication cycles were repeated three times. The samples were incubated for 1 h at −40 °C and centrifuged at 12,000 rpm for 15 min at 4 °C. The resulting supernatant was transferred to a fresh glass vial for analysis.

LC-MS/MS analyses were performed using a UHPLC system (Vanquish; Thermo Fisher Scientific) with a UPLC BEH Amide column (2.1 mm × 100 mm, 1.7 μm) coupled to a Q Exactive HFX mass spectrometer (Orbitrap MS; Thermo Fisher Scientific). The mobile phase consisted of 25 mmol/L ammonium acetate and 25 mmol/L ammonia hydroxide in water (pH 9.75) (A) and acetonitrile (B). The auto-sampler temperature was 4 °C, and the injection volume was 2 μl. The QE HFX mass spectrometer was applied to acquire MS/MS spectra using the information-dependent acquisition mode with acquisition software (Xcalibur; Thermo Fisher Scientific). In this mode, the acquisition software continuously evaluated the full MS spectrum. The emergency severity index source conditions were set as follows: sheath gas flow rate 30 Arb, Aux gas flow rate 25 Arb, capillary temperature 350 °C, full MS resolution 60,000, MS/MS resolution = 7500, collision energy 10/30/60 in normalized collisional energy mode, and spray voltage 3.6 kV (positive) or −3.2 kV (negative).

The raw data were converted to the mzXML format using ProteoWizard and processed with an in-house program (developed using R and based on XCMS) for peak detection, extraction, alignment, and integration. Then, an in-house MS2 database (BiotreeDB) was applied for metabolite annotation. The cutoff for annotation was set at 0.3.

**RNA isolation, RNA-seq, and data processing.** Total RNA was extracted from tissues using Trizol reagent (Invitrogen, Carlsbad, CA, USA). Oligo (dT)-attached magnetic beads were used to purify mRNA. Purified mRNA was fragmented into small pieces with fragment buffer at the appropriate temperature. Then, first-strand cDNA was generated using random hexamer-primed reverse transcription, followed by second-strand cDNA synthesis. Afterward, A-Tailing Mix and RNA Index Adapters were added by incubating to end repair. The cDNA fragments obtained in the previous step were amplified by PCR, and products were purified using Ampure XP beads and then dissolved in an elution buffer solution. The product was validated on the Agilent Technologies 2100 bioanalyzer for quality control. The double-stranded PCR products from the previous step were heated, denatured, and circularized by the splint oligo sequence to obtain the final library. Single-strand circular DNA was formatted as the final library. The final library was amplified with phi29 to generate DNA nanoballs containing more than 300 copies of one molecule. DNA nanoballs were loaded into the patterned nanoarray, and

paired-end 50-base-pair reads were generated on the BGIseq500 platform (BGI, Shenzhen, China).

The sequencing data were filtered with SOAPnuke (v1.5.2)[41] by: (1) removing reads containing sequencing adapters, (2) removing reads with a low-quality base ratio (base quality ≤ 5) >20, and (3) removing reads with an unknown base (N' base) ratio >5%. Clean reads were obtained and stored in FASTQ format and were mapped to the reference genome using HISAT2 (v2.0.4)[42]. Bowtie2 (v2.2.5)[43] was applied to align the clean reads to the reference coding gene set, and then the gene expression level was calculated by RSEM (v1.2.12)[44]. The heatmap was drawn by pheatmap (v1.0.8) according to the gene expression in different samples. Differential expression analysis was performed using the DESeq2(v1.4.5)[45] with Q < 0.05. To gain insight into phenotypic changes, GO (http://www.geneontology.org/) and KEGG (https://www.kegg.jp/) enrichment analysis of annotated differentially expressed genes were performed by Phyper (https://en.wikipedia.org/wiki/Hypergeometric distribution) based on the hypergeometric test. The significant level of terms and pathways were corrected using the Q value with a rigorous threshold (<0.05) by the Bonferroni test.

**Reverse transcription-quantitative PCR**. RNA was treated with DNase, and 1 µg RNA was used for reverse transcription. cDNA diluted 10× was used for reverse transcription-quantitative PCR (RT-qPCR). RT-qPCR was performed using the Light-Cycler system (Roche Diagnostics GmbH, Rotkreuz, Switzerland) and a qPCR Supermix (Vzayme, Nanjing, China) with the indicated primers. An average of at least three technical repeats was used for each biological data point. The *ropB* gene was used as the reference gene, and the following primers were used: *ropB*-F, CTGCGCGAAGAAATCGAAGG, *ropB*-R: TTTCGCCAACGGAACGGATA and *PncA*-F: TGATCGCCAGCCAAGACT, *PncA*-R: AGCATCCAGCACCGTGAA.

**Western blotting**. Bacteria (10⁹ cfu) were lysed in 200 µl lysis buffer (from the protein purification kit). Samples of 20 µl were loaded onto 12.5% Bis-Tris poly-acrylamide gels and transferred onto PVDF membrane (Millipore, Billerica, MA, USA) by electroblotting. The membranes were blocked in Tris-buffered saline and 0.5% Tween containing 5% skimmed milk powder (OXOID, Basingstoke, UK) for 1 h at room temperature and incubated with primary antibody, anti-his (M20001; Abmart, Shanghai, China) with 1:1000 dilution (500 µg/ml) overnight at 4 °C. The membranes were incubated with peroxidase-conjugated secondary antibody (Bio-Rad, Hercules, CA, USA) for 1 h at room temperature. The bands were visualized using Millipore's enhanced chemiluminescence with the Amersham Imager 600 detection system (GE, Boston, USA).

**16 S rRNA sequencing and analysis**
*Genomic DNA extraction*. Microbial community DNA was extracted using a MagPure Stool DNA KF kit B (Magen, Guangzhou, China). DNA was quantified with a Qubit Fluorometer using a Qubit dsDNA BR Assay kit (Invitrogen), and the quality was assessed by running an aliquot on a 1% agarose gel.

*Library construction*. Variable regions V3–V4 of the bacterial 16S rRNA gene were amplified with the degenerate PCR primers 341F (5′-ACTCCTACGGGAGG-CAGCAG-3′) and 806R (5′- GGACTACHVGGGTWTCTAAT-3′). Both forward and reverse primers were tagged with Illumina adapter, pad, and linker sequences. PCR enrichment was performed in a 50-µl reaction containing 30 ng template, fusion PCR primer, and PCR master mix. PCR cycling conditions are as follows: 94 °C for 3 min, 30 cycles of 94 °C for 30 s, 56 °C for 45 s, 72 °C for 45 s, and a final extension for 10 min at 72 °C. The PCR products were purified with Ampure XP beads and eluted in Elution buffer. Libraries were qualified with the Agilent 2100 bioanalyzer (Agilent, Santa Clara, CA, USA). The validated libraries were used for sequencing on an Illumina MiSeq platform (BGI) following the standard pipelines of Illumina, and 2 × 300 bp paired-end reads were generated.

*Data processing*. First, low-quality data were removed from the original sequencing data by the window method with Readfq v8 (https://github.com/cjfields/readfq). Joint pollution reads and n-containing reads were removed, then low-complexity reads were processed. Samples were distinguished based on barcode and primer. FLASH software[46] (fast length adjustment of short reads, v1.2.11) was used for assembling. Using overlapping relationships, pairs of double-end sequencing reads were assembled into a sequence with high area tags. Effective tags were produced by the UCHIME algorithm and clustered into operational taxonomic units (OTUs) using USEARCH[47] (v7.0.1090) software. According to the mothur method and Greengenes database, taxonomic information was annotated with representative sequences from OTUs. Phy tools[48] and R software (v3.4.1) were used to perform an unweighted pair group method with arithmetic mean clustering analysis based on Bray–Curtis weighted Unifrac and unweighted Unifrac distance matrices. The ade4[49] package of R (v3.4.1) was used to perform an OTU PCA analysis. The RDP classifier Bayesian algorithm was used to classify the OTU representative sequences. The community composition of individual samples was counted at the species level of the phylum, order, family, and genus, and the histogram of species abundance was performed using the ggplot2 package of R. Alpha diversity statistics were analyzed using the software motor (v1.31.2). Beta diversity was analyzed by QIIME (v1.80)[50]. Finally, the ggplot2 package of R was used for box plots of alpha and beta diversity.

**Oil Red staining and H&E staining**. Hepatic tissue was cut into small pieces, fixed in 4% paraformaldehyde for 4 h, and embedded in OCT (Leica Camera AG, Wetzlar, Germany). Frozen sections (8 µm thickness) were made using a cryostat, and the samples were fixed with 4% paraformaldehyde for an additional 30 min. The slides were washed in distilled water and stained with Oil Red O for 15 min. Next, the slides were counterstained with hematoxylin for 10 s to identify the nuclei. For H&E staining, the slides were first stained in hematoxylin for 3–8 min and then counterstained with eosin for 1–3 min. The histological images were acquired with a light microscope (Olympus, Tokyo, Japan).

**NAD⁺ detection**. NAD⁺ detection was carried out using the EnzyChrom TM NAD⁺/NADH⁺ Assay Kit (E2ND-100) from BioAssay Systems (Hayward, CA, USA). Protein concentration was used to normalize the NAD⁺ content.

**ATP detection**. For the measurement of ATP level, 100-mg liver samples were lysed in 1 ml lysis buffer provided by the ATP Assay Kit (S0026) from Beyotime (Jiangsu, China). Liver ATP levels were evaluated by luciferase activity, as shown in the standard protocol provided by the ATP Assay Kit.

**Triglyceride detection**. Liver triglycerides were assayed using a triglyceride assay kit (E1025; Applygen Technologies, Beijing, China).

**Statistics and reproducibility**. All data were analyzed using GraphPad Prism 8 (Graphpad Software Inc.). Differences between the two groups were evaluated using Student's *t*-test with a two-tailed distribution. All data are presented as the mean ± standard deviation (SD) from at least three independent experiments performed in triplicate. Sample sizes were selected without performing statistical tests. No data were excluded when conducting the final statistical analysis. *$p < 0.05$, **$p ≤ 0.01$, ***$p ≤ 0.001$, and ****$p < 0.0001$ unless stated otherwise.

**Reporting summary**. Further information on research design is available in the Nature Portfolio Reporting Summary linked to this article.

## Data availability
The source data underlying Figs. 2a, 2c–f, 3d, 4a, 4c–e, and 5a–d, Supplementary Figs. S2c, S3d, S5a–d, S6a, S6c, d, S7c, and S8a, b are provided as Supplementary Data 2. The RNA-seq raw sequencing data files generated in this study are available in the NCBI's Sequence Read Archive (SRA) under BioProject accession number PRJNA780659. The original 16 S rRNA sequence data are available at the NCBI by accession number PRJNA780391. All other data are available from the corresponding author.

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

## Acknowledgements

This work was supported by grants from the National Key Research and Development Program of China (2020YFC2002800), the National Natural Science Foundation of China (82271593), and the Fundamental Research Funds for the Central University (22120210584).

## Author contributions

L.H.L and F.S.Y designed the study; F.S.Y., Y.S.S., W.H., and G.L.L analyzed data; L.H.L supervised the entire project, and all authors approved the final version of the manuscript.

## Competing interests

The authors declare no competing interests.
