## [Peer Review File · Communications Biology]

Reviewers' comments:

Reviewer #1 (Remarks to the Author):

The manuscript "PncA from bacteria improves diet-induced NAFLD by enabling the transition from NAM to NA in mice" examines the role of bacterial pncA in host NAD metabolism. The authors first demonstrate that an analog of a common anti-tuberculosis drug PCN, which is an inhibitor of bacterial pncA, change gut microbiome composition and function and reduces host NAD levels. Then, the authors delete or overexpress pncA in E.coli and show that repopulation of antibiotics-treated mice with the overexpressing strain results in slight attenuation of liver NAD levels decline and of body weight loss in MCD diet model of NAFLD, when combined with NAM supplementation. They then decided to overexpress bacterial pncA directly in the mouse liver and demonstrate much more dramatic effects of elevation of liver NAD and protection from NAFLD.

Overall, this is an interesting work and a direct follow up on a previous study that demonstrated the importance of bacterial pncA for host NAD metabolism (ref 11). Although not conceptually novel, providing experimental evidence that manipulating pncA activity has therapeutic benefits is new and important.

The work has some important technical caveats detailed below, which need to be addressed. Although the impressive results with adenovirus-mediated overexpression of pncA in the liver provide important evidence on the therapeutic potential of activating the deamidated NAD biosynthesis pathway, translational potential of such gene therapy in the clinic is still limited. On the other hand, pncA-overexpressing probiotics seem to be more therapeutically feasible approach.

Technically, the work lacks sufficient experimental details throughout, and the authors should provide such details to enable replication by others.

Specific points:

1. The title is a bit misleading as it implies that PncA comes from bacteria. However, when the authors modulated pncA activity in bacteria, the effects were only marginal. The most dramatic result was by overexpression of this bacterial enzyme in the liver using adenoviral vector. Thus, I would propose to modify the title:

"Bacterial PncA improves diet-induced NAFLD by enabling the transition from NAM to NA in mice".

2. The methods for phylogenetic analysis of bacterial NAD biosynthesis pathways in different organisms (Fig S1) is not described at all. The authors should describe in detail the databases and analytical procedures.

3. Fig. 2A. number of replicates is not indicated. Individual replicates should be shown in addition to means as in C-F.

4. FigS2- the resolution is too low and the labels are completely illegible. I suggest breaking this figure up and use larger fonts to improve quality.

5. Fig2. Experimental scheme should be added (timing of PNC, NAM additions).

6. Fig2. Details of bioinformatic analysis of 16S sequencing should be described in Methods

7. In the PCN experiment, NAD, NAM and NA content in bacteria (feces) should also be measured
8. The increased richness of bacteria after PCN is interesting. The authors should compare the bacterial composition in relation to their NAD pathway enzymes before and after PCN treatment. Are PncA-dependent bacteria depleted and de-novo-pathway-dependent bacteria enriched after PCN?
9. Fig2E: one would expect that inhibition of pncA by PCN should result in accumulation of NAM in the colon and reduction of NA (which should be measured and shown). However, the result is opposite (reduction of NAM)? The authors should explain this unexpected result instead of describing it as "reduced the utilization efficiency of NAM in the liver and intestine of the host, resulting in lower NAD+ levels." To claim reduced utilization efficiency of NAM, they need to use isotopically labeled NAM and measure its incorporation into liver, colon NAD.
10. The authors attempted to prove that the effects of PCN on host gene expression and NAD levels are mediated by microbiota using in-vitro treatment of cell lines. This is insufficient as in-vitro lines do not reflect all in-vivo complexity and do not represent the variety of tissues that might be affected by the drug. A better control would be to treat with PCN mice with depleted microbiota (either germ-free or antibiotics-treated).
11. Fig S3C, S3D Y-axis should start from zero
12. Fig. S4 The description of methods for construction of pncA-overexpressing E.coli is missing (vector, inducible or constitutive). Also, overexpression should be demonstrated on the protein level in addition to RNA level (using tagged pncA). The RNA expression should include "no RT"-control to rule out the possibility that the signal comes from the amplification of the plasmid DNA copurified with RNA.
13. Lines 226, 277: "metabolome sequencing" should be changed to "metabolome analysis" since LC-MS is not a sequencing technology. Same for Fig 3A schematics
14. Fig3. It is crucial to repeat all the experiments including an additional group of WT E.coli with empty vector control. This is necessary to judge the degree of functional pncA overexpression compared to the activity of this enzyme in WT E.coli. It is possible that the underwhelming results showed for pncA overexpression in bacteria stem from some technical overexpression problem. Comparison of NA levels in the supernatant between vector-only and pncA-overexpressing bacteria should indicate whether overexpression was functionally successful. Only after showing very strong increase in NA production, should the authors proceed to the in-vivo experiments, where they will need to measure NA in the feces to make sure that NA continues to be overproduced by overexpressing bacteria compared with vector-only E.coli at several time points along the colonization experiment.
15. Fig 3B. What is the interpretation of decrease in NAD, vit B2, asparagine in the culture medium of pncA KO E.coli compared with overexpression? Is there any effect on growth rate?
16. Fig3F,G the labels are confusing: Why AAV-vector is shown in this experiment focused on overexpression of pncA in E.coli? Is "pncA-Vector" the overexpressing strain or control strain?

17. For the experiments on Fig.3 , the phenotypic endpoint shown is animal weight but for the experiment on Fig4 (overexpression of pncA in the liver), weight curves are not shown but instead liver histology and triglycerides are shown.

The authors should show all three endpoints for both types of experiments.

18. Line 312 "reappear" should be changed to "recapitulate"

19. Metabolomics data should be provided as supplementary Excel tables.

Reviewer #2 (Remarks to the Author):

This manuscript by Liu et al. adds to the role of gut-microbiota PncA, a biosynthetic enzyme salvaging nicotinamide and re-routing this NAD precursor into the deamidating salvage pathway, which, when engaged (e.g., with supplements), is more efficient than the amidated one in boosting mammalian NAD metabolism and can sustain cancer resistance in NAMPi-targeted therapies.

Here, the authors deliver new data that elevates the importance of gut-microbial PncA to a key role a) for the well-being of the microbiome itself, b) for supporting host NAD physiological levels, c) in ameliorating NAFLD condition in mice indicating the bacterial PncA could be a promising target to treat this and other NAD-deficient diseases.

Their approach mainly involved the use of PCN, a potent PncA inhibitor, as a molecular probe in their experimental designs.

Overall, the paper is well-written and presents solid results that are in the main interest of this field. I have, however, several issues that must be duly addressed.

MAJOR concerns

1. Some figures in this manuscript are hard to see and interpret. For example, in Fig. 4, panel F words are hard to read, so do the Y-axis Title and all scale numbers. Text is also not clear in Fig. 3 and Fig.

1. In the latter, some abbreviations are too small, and color contrast is terrible: a better combination of colors for the text abbreviations and circles' background should be used. The situation is much worse for Supplemental Figures where, in addition to the poor resolution, most figures are crowded with too many (and small) panels (Supplementary Figures 2, 4, and 5). I don't know whether the resolution has been lost in the pdf assembly or the original figures had poor resolution since their creation. In any event, authors should produce figures at 300 d.p.i. minimum, and enlarge panels and text labels to improve readability.

2A. Most of the author's claims derives from using pyrazinecarbonitrile as an inhibitor of microbial-derived PncA. PCN was discovered as an inhibitor of Mtb PncA, as adequately cited in the text. How can the authors support the notion (implicit in the text) that PCN is broadly active on non-Mtb-bacteria?. If already reported in the literature, please add those references.

2B. One claim, in particular, is that PncA inhibitor PCN disrupts microbiome homeostasis, as seen with a significant increase in bacterial expansion and richness compared to control mice upon PCN treatment. Being PncA a bacterial enzyme contributing to the synthesis of an essential cofactor, I found it a bit unexpected overall. How can the authors rule out that microbiome PCN-induced homeostasis disruption is dependent, in full or partially, by general toxicity and/or other side-activities of PCN? In this regard, it should be noted that PCN is an irreversible inhibitor of Mtb PncA.

2C. Surprisingly, the authors do not mention in the results or the methods or figure legends what concentration of PCN and nicotinamide was used in their key in vitro or in vivo experiments. This is crucial for interpreting the results.

3. The authors refer to the first-line tuberculosis drug pyrazinamide (chemically analog of PCN) as "a bacterial NAD synthesis" inhibitor (Line 38) and state that pyrazinamide targets PncA (Line 86). This is

incorrect!. On the contrary, PncA is required to convert the prodrug pyrazinamide into pyrazinoic acid, which then exerts its antibacterial activity hitting other target(s). It is true, though, that PZA can synergize with NAD biosynthesis inhibitors (PMID: 33584600).

4. In Fig. S1 a genomic reconstruction of NAD metabolism is presented for various bacteria, some of which were selected for in vitro and in vivo experiments due to their peculiar repertoire of NAD synthesis pathways. What bioinformatics tools or resources were used to make the calls for the presence or absence of a viable NAD-related enzyme?. In particular, PncA, could be at times of difficult assignment, being confused with other amidases of isochorismatase family, and only genome context analysis can yield reliable assignment. Also, how is it possible to have genes of eukaryotic origin like QSN1?. QSN1 is better known as NADS, NAD synthetase, equivalent to nadE in bacteria. Even if this is a duplication error for the same functional role, how can it possibly be that in some organisms, nadE is present and qsn1 is not: this makes no sense.

The author should perform a better bioinformatics reconstruction and report the method in the experimental results section, and briefly mention it in the results.

What criterium was used to select the bacterial species?. If these all belong to microbiota, it should be stated.

5. Suriwhat is the concentration of PCN and Nam used in the experiments? (see Fig. 2

Other comments (in order of appearance in the text)

4. Line 41: when referring to the functional role of PncA, the authors should use either nicotinamidase alone (the actual physiological role) or nicotinamidase/pyrazinamidase and NOT pyrazinamidase/nicotinamidase. Please correct here and in other instances (e.g., Line 76, etc.).

Lines 64-67: please provide REF(s) for these statements.

5. Lines 105-119 (Fig. 1 title and legend). Figure 1 legend is missing the explanation of all abbreviations, enzymes of metabolites. Some abbreviations are incorrect and should be fixed: Try with Trp; NMAM with MNAM; nam 1-2 is probably Nma1-2 (yeast-like NMNAT), better referred to as Nmnat1-2 for consistency. In figure 1, it is shown that NAM is converted to methylNAM; in what organism of the gut microbiota NNMT enzyme is present?. In the gut microbiota section, two conversions regarding NAMN to NMN by NMN synthetase NadE* or NMNS (PMID: 19204287) and NMN to NaMN by NMN deamidase PncC (PMID: 21953451). These two reactions are also unique to gut microbiota. I recommend an accurate reading of recent and/or comprehensive papers on the genomics and enzymology of NAD biosynthesis to avoid these mistakes (e.g. book chapter In Comprehensive Natural Products II Chemistry and Biology; Mander, L., Lui, H.-W., Eds.; Elsevier: Oxford, 2010; volume 7, pp.213-257 and many others).

6. Supplementary Figure 1. This figure, as stated in the title, only pertains to bacteria. Thus, all eukaryotic functional roles should not be included (e.g. Qns1, NPT1, etc) as they do not exist in bacteria.

7. Line 131. If I understood correctly, only PCN inhibitor was used. Thus, replace "inhibitors" with "inhibitor". Or, even better, change the title into PncA inhibitor PCN disrupts....

8. line 135. After antibiotic, resume continuous reading. Do not go head here.

9. Fig. 2A. The authors performed growth experiments of selected bacteria and talked about growth rates. Instead, they measured the final Optical Density after 24 hours, when bacteria could have reached similar densities with different rates. This experiment should be better performed by monitoring the kinetics of growth, i.e., OD/time, and then the results interpreted accordingly.

10. Line 168-9. Fig. 2F does not match what is stated in the text.

10. How do the authors explain the expansion of *Bifidobacterium longum*, a bacterium that apparently contains pncA, upon PCN treatment?

11. Line 225. "The bacteria were cultured for the indicated period". Indicated where?. At what concentration Nam was used, and at what stage were bacteria collected?

12. Line 339. Add "salvage of" before "NA and NAM".

Response to Reviewers

Reviewer 1:

1. The title is a bit misleading as it implies that PncA comes from bacteria. However, when the authors modulated pncA activity in bacteria, the effects were only marginal. The most dramatic result was by over-expression of this bacterial enzyme in the liver using adenoviral vector. Thus, I would propose to modify the title:

“Bacterial PncA improves diet-induced NAFLD by enabling the transition from NAM to NA in mice”.

Response: We have changed the title to “Bacterial PncA improves diet-induced NAFLD by enabling the transition from NAM to NA in mice”, which more accurately expresses the subject of our study.

2. The methods for phylogenetic analysis of bacterial NAD biosynthesis pathways in different organisms (Fig S1) is not described at all. The authors should describe in detail the databases and analytical procedures.

Response: We have added the analytical procedures of Fig. S1 (now Fig. S2) to the Methods.

3. Fig. 2A. number of replicates is not indicated. Individual replicates should be shown in addition to means as in C-F.

Response: We reanalyzed the growth rate of bacteria by monitoring the growth kinetics, which more accurately reflected the effect of PCN on the growth of bacteria. The number of replicates is indicated in Fig. 2.

4. FigS2- the resolution is too low and the labels are completely illegible. I suggest breaking this figure up and use larger fonts to improve quality.

Response: We have divided Fig. S2, adjusted the layout of the image and made the font larger (Now Figs. S3 and S4)

5. Fig2. Experimental scheme should be added (timing of PNC, NAM additions).

Response: PNC and NAM were added every day, which is explained in detail in the Methods, so we think there is no need to clarify the experimental process.

6. Fig2. Details of bioinformatic analysis of 16S sequencing should be described in Methods

Response: We have added details about 16S sequencing and analysis to the Methods.

7. In the PCN experiment, NAD, NAM and NA content in bacteria (feces) should also be measured

Response: We repeated the PCN experiment, collected the feces of mice and measured the content of NAM, NA and NAD⁺. The amount of NAM and NA is shown in Fig. S3D, while NAD⁺ was not detected in feces.

8. The increased richness of bacteria after PCN is interesting. The authors should compare the bacterial composition in relation to their NAD pathway enzymes before and after PCN treatment. Are PncA-dependent bacteria depleted and de-novo-pathway-dependent bacteria enriched after PCN?

Response: We have previously analyzed the potential relationship between enriched bacteria and NAD⁺ synthesis pathway. However, there was no obvious evidence that showed a decrease in PncA-dependent bacteria and enrichment of *de-novo*-pathway-dependent bacteria after treatment with PCN. We think it might have resulted from the complexity of the gut microbiota and close interaction between them. The complex dependence of gut bacteria on each other makes it difficult to judge bacterial fluctuations by changes in one factor.

9. Fig2E: one would expect that inhibition of pncA by PCN should result in accumulation of NAM in the colon and reduction of NA (which should be measured and shown). However, the result is opposite (reduction of NAM)? The authors should explain this unexpected result instead of describing it as “reduced the utilization efficiency of NAM in the liver and intestine of the host, resulting in lower NAD⁺ levels.” To claim reduced utilization efficiency of NAM, they need to use isotopically labeled NAM and measure its incorporation into liver, colon NAD.

Response: I am not sure if we understood your points correctly, because we did not show a reduction of NAM after PCN treatment (Fig. 2E). The “NAM, NAM+PCN” in Fig. 2 was the group names, and the values represent NAD⁺ content of each group. Please correct me if we have not understood correctly.

10. The authors attempted to prove that the effects of PCN on host gene expression and NAD levels are mediated by microbiota using in-vitro treatment of cell lines. This is insufficient as in-vitro lines do not reflect all in-vivo complexity and do not represent the variety of tissues that might be affected by the drug. A better control would be to treat with PCN mice with depleted microbiota (either germ-free or antibiotics-treated).

Response: Yes, germ-free or antibiotic-treated mice were indeed better controls. Because of the limitations of our laboratory conditions, we chose antibiotic-treated mice as the model. Fig. S5D shows that no change in NAD⁺ was observed after PCN treatment of the antibiotic-treated mice.

11. Fig S3C, S3D Y-axis should start from zero

Response: We have modified the Y-axis of Fig. S3C, S3D (now Fig. S5).

12. Fig. S4 The description of methods for construction of *pncA*-overexpressing *E. coli* is missing (vector, inducible or constitutive). Also, overexpression should be demonstrated on the protein level in addition to RNA level (using tagged *pncA*). The RNA expression should include “no RT”-control to rule out the possibility that the signal comes from the amplification of the plasmid DNA copurified with RNA.

Response: We have added details of the methods for construction of PncA-overexpressing *E. coli*. The results of western blotting and modified qPCR of PncA in PncA-overexpressing *E. coli* are shown in Fig. S6.

13. Lines 226, 277: “metabolome sequencing” should be changed to “metabolome analysis” since LC-MS is not a sequencing technology. Same for Fig 3A schematics

Response: We have corrected the text and figure accordingly (Lines 175, 225).

14. Fig3. It is crucial to repeat all the experiments including an additional group of WT *E. coli* with empty vector control. This is necessary to judge the degree of functional *pncA* overexpression compared to the activity of this enzyme in WT *E. coli*. It is possible that the underwhelming results showed for *pncA* overexpression in bacteria stem from some technical overexpression problem. Comparison of NA levels in the supernatant between vector-only and *pncA*-overexpressing bacteria should indicate whether overexpression was functionally successful.

Only after showing very strong increase in NA production, should the authors proceed to the in-vivo experiments, where they will need to measure NA in the feces to make sure that NA continues to be overproduced by overexpressing bacteria compared with vector-only *E. coli* at several time points along the colonization experiment.

Response: A WT *E. coli* group with vector plasmid would indeed be a better control, as there were more than one variable factors between PncA-KO and PncA-OE *E. coli*. We repeated the bacteria and mouse experiments. Fig. 3B showed overproduction of NA in the supernatant of PncA-OE *E. coli* compared with WT *E. coli*. Besides, we verified that PncA was functionally successful in the *in vitro* enzyme activity experiment. Fig. S6C shows that PncA activity was hundreds of times greater than in *E. coli* with only empty vector.

However, for the mouse experiment, Fig. 6D shows that there was no overproduction of NA in the feces in the colonization experiment. There were two or more possible reasons. First, antibiotic treatment dramatically changed the homeostasis of the gut flora and colon function, and colonization by *E. coli* did not recover the original function of the gut

microbiome. Second, the *in vitro* enzyme activity of PncA-OE *E. coli* may not indicate that the bacteria have such activity *in vivo* because the environment changed dramatically after colonization. As you stated, we need to make sure that NA continues to be overproduced by overexpressing bacteria compared with vector-only *E. coli*. We think that germ-free mice would be a good model for our research, but we were limited by our laboratory conditions and the high price of germ-free mice, so we used the AAV method.

15. Fig 3B. What is the interpretation of decrease in NAD, vit B2, asparagine in the culture medium of pncA KO *E.coli* compared with overexpression? Is there any effect on growth rate?

Response: Some other factors changed after PncA KO; however, we focused on the influence of PncA on the transition of NAM to NA. PncA may have influenced the metabolic process, and further changed the concentration of factors related to metabolism.

16. Fig3F,G the labels are confusing: Why AAV-vector is shown in this experiment focused on overexpression of pncA in *E.coli*?

Is “pncA-Vector” the overexpressing strain or control strain?

Response: We confused the group names of the two experiments, and have changed the X-axis of Fig. 4C–E to coincide with that of Fig. 3D.

17. For the experiments on Fig.3, the phenotypic endpoint shown is animal weight but for the experiment on Fig4 (overexpression of pncA in the liver), weight curves are not shown but instead liver histology and triglycerides are shown.

The authors should show all three endpoints for both types of experiments.

Response: We have supplemented the relevant experiments. Changes in body weight of the mice (AAV experiment) are shown in Fig. S8, and liver histology and triglycerides in the bacterial experiments are shown in Fig. 4B.

18. Line 312 “reappear” should be changed to “recapitulate”.

Response: We have changed this accordingly (Line 238).

19. Metabolomics data should be provided as supplementary Excel tables.

Response: Metabolomics data have been attached as supplementary files (Line 234).

Reviewer 2:

1. Some figures in this manuscript are hard to see and interpret. For example, in Fig. 4, panel F words are hard to read, so do the Y-axis Title and all scale numbers. Text is also not clear in Fig. 3 and Fig. 1. In the latter, some abbreviations are too small, and color

contrast is terrible: a better combination of colors for the text abbreviations and circles' background should be used. The situation is much worse for Supplemental Figures where, in addition to the poor resolution, most figures are crowded with too many (and small) panels (Supplementary Figures 2, 4, and 5). I don't know whether the resolution has been lost in the pdf assembly or the original figures had poor resolution since their creation. In any event, authors should produce figures at 300 d.p.i. minimum, and enlarge panels and text labels to improve readability.

Response: We acknowledge the low quality of our figures. The original figures had good resolution. However, during the assembly and compression process, most images were not as clear as they should have been. We have adjusted the layout and clarity of all figures to improve their readability.

2A. Most of the author's claims derives from using pyrazinecarbonitrile as an inhibitor of microbial-derived PncA. PCN was discovered as an inhibitor of Mtb PncA, as adequately cited in the text. How can the authors support the notion (implicit in the text) that PCN is broadly active on non-Mtb-bacteria? If already reported in the literature, please add those references.

Response: We have not found any studies that have shown that PCN is and inhibitor of Mtb and all microbial-derived PncA. However, the active catalytic sites of Mtb PncA are conserved in almost all bacteria (Fig. S1), so we have good reason to believe that PCN also functions in other bacteria. We purified PncA protein of *E. coli* and Fig. 1C confirms this conclusion. Besides, PCN had a significant influence of the growth of bacteria that only have the Preiss–Handler pathway for NAD synthesis.

2B. One claim, in particular, is that PncA inhibitor PCN disrupts microbiome homeostasis, as seen with a significant increase in bacterial expansion and richness compared to control mice upon PCN treatment. Being PncA a bacterial enzyme contributing to the synthesis of an essential cofactor, I found it a bit unexpected overall. How can the authors rule out that microbiome PCN-induced homeostasis disruption is dependent, in full or partially, by general toxicity and/or other side-activities of PCN? In this regard, it should be noted that PCN is an irreversible inhibitor of Mtb PncA.

Response: Our results for PCN treatment in mice were unexpected; However, the same results were obtained when we repeated that experiment. The effects of drugs on gut microbiota are further complicated by the close inter-relationship between gut bacteria. A decline in one bacterium may lead to an increase in the number of bacteria whose growth is inhibited by that bacterium, which may be one of the reasons that led to the increase of bacteria.

As for the potential general toxicity of PCN, we used antibiotic-treated mice as the new control group, and we found that PCN had no effect on liver and colon NAD⁺ after antibiotic treatment (Fig. S5D). This indicates that the effect of PCN on the host is largely dependent on the microbiome rather than its side effects.

2C. Surprisingly, the authors do not mention in the results or the methods or figure legends what concentration of PCN and nicotinamide was used in their key in vitro or in vivo experiments. This is crucial for interpreting the results.

Response: This section was missing from the Methods. We have added the detailed procedures for the PCN experiment to the Methods.

3. The authors refer to the first-line tuberculosis drug pyrazinamide (chemically analog of PCN) as “a bacterial NAD synthesis” inhibitor (Line 38) and state that pyrazinamide targets PncA (Line 86). This is incorrect!. On the contrary, PncA is required to convert the prodrug pyrazinamide into pyrazinoic acid, which then exerts its antibacterial activity hitting other target(s). It is true, though, that PZA can synergize with NAD biosynthesis inhibitors (PMID:33584600).

Response: We acknowledge that our statement about PCN function was inappropriate. We have modified it accordingly.

4. In Fig. S1 a genomic reconstruction of NAD metabolism is presented for various bacteria, some of which were selected for in vitro and in vivo experiments due to their peculiar repertoire of NAD synthesis pathways. What bioinformatics tools or resources were used to make the calls for the presence or absence of a viable NAD-related enzyme? In particular, PncA, could be at times of difficult assignment, being confused with other amidases of isochorismatase family, and only genome context analysis can yield reliable assignment.

Also, how is it possible to have genes of eukaryotic origin like QSN1?. QSN1 is better known as NADS, NAD synthetase, equivalent to nadE in bacteria. Even if this is a duplication error for the same functional role, how can it possibly be that in some organisms, nadE is present and qsn1 is not: this makes no sense. The author should perform a better bioinformatics reconstruction and report the method in the experimental results section, and briefly mention it in the results. What criterium was used to select the bacterial species? If these all belong to microbiota, it should be stated.

Response: The method that we used to screen NAD-related enzymes is cited in PMID: 32130883, and the detailed bioinformatics analysis of NAD-related enzymes has been

added to the Methods. As for PncA screening in bacteria, to eliminate the interference of other members in isochorismatase, the DALI (protein structure alignment server) tool was used to verify candidate PncA proteins by aligning with known PncA of *E. coli*, because PncA has a conserved nucleophilic cysteine that is absent in isochorismatase in other bacteria.

We confused QSN1 and nadE in Fig. S1. nadE is the name of NAD synthetase in bacteria. To explain why QNS1 was also present in Fig. S1, to maximize the detection of NAD synthetase, we used QNS1 of *Saccharomyces cerevisiae* and nadE of *E. coli* to screen nadE in bacteria, because QNS1 is a homolog of nadE, however, the length of QNS1 is about three times that of nadE, which is different from nadE in structure. And that's why nadE is present and QNS1 is not in some bacteria. It was not really not appropriate here, so we have deleted QNS1 from Fig. S1. The bacteria analyzed in Fig.S1 (now Fig. S2) all belong to mammalian microbiota; mostly are gut microbiota and some are pathological bacteria. We have added this statement to the Methods.

5. What is the concentration of PCN and Nam used in the experiments? (See Fig. 2 Other comments (in order of appearance in the text))

Response: We have added details about the PCN experiment in bacteria and mice to the Methods.

6. Line 41: when referring to the functional role of PncA, the authors should use either nicotinamidase alone (the actual physiological role) or nicotinamidase/pyrazinamidase and NOT pyrazinamidase/ nicotinamidase. Please correct here and in other instances (e.g., Line 76, etc.). Lines 64-67: please provide REF(s) for these statements.

Response: We have revised our statement about the function of PncA in the Introduction and used the abbreviation in the main text. A reference has been added for Lines 67.

7. Lines 105-119 (Fig. 1 title and legend). Figure 1 legend is missing the explanation of all abbreviations, enzymes of metabolites. Some abbreviations are incorrect and should be fixed: Try with Trp; NMAM with MNAM; nam 1-2 is probably Nma1-2 (yeast-like NMNAT), better referred to as Nmnat1-2 for consistency. In figure 1, it is shown that NAM is converted to methylNAM; in what organism of the gut microbiota NNMT enzyme is present?. In the gut microbiota section, two conversions regarding NAMN to NMN by NMN synthetase NadE* or NMNS (PMID: 19204287) and NMN to NaMN by NMN deamidase PncC (PMID: 21953451). These two reactions are also unique to gut microbiota. I recommend an accurate reading of recent and/or comprehensive papers on the genomics and enzymology of NAD biosynthesis to avoid these mistakes (e.g. book chapter In Comprehensive Natural Products II Chemistry and Biology; Mander, L., Lui, H.-W., Eds.;

Elsevier: Oxford, 2010; volume 7, pp.213–257 and many others).

Response: We have modified Fig. 1 and related abbreviations (Lines 721-740).

Firstly, we exclude all fungal gene names, because our focus was on bacteria. Secondly, there are several candidate nicotinamide N-methyltransferases of bacteria in the Uniprot database without experimental evidence, such as M2X6A7 and A0A328N1E2, but we decided to exclude the NNMT pathway in bacteria because of the lack of experimental evidence. Finally, we have added the pathways that were catalyzed by NadE and PncC to Fig. 1; however, ftNadE seems to exist in several bacteria because of its specific protein structure.

8. Supplementary Figure 1. This figure, as stated in the title, only pertains to bacteria. Thus, all eukaryotic functional roles should not be included (e.g. Qns1, NPT1, etc) as they do not exist in bacteria.

Response: The eukaryotic enzymes were indeed not appropriate considering that our research focused on bacteria. Therefore, we have excluded the eukaryotic genes from Fig. S1. Thanks.

9. Line 131. If I understood correctly, only PCN inhibitor was used. Thus, replace “inhibitors” with “inhibitor”. Or, even better, change the title into PncA inhibitor PCN disrupts....

Response: We have modified “inhibitors” to “inhibitor” (Line 104).

10. line 135. After antibiotic, resume continuous reading. Do not go head here.

Response: We have modified the corresponding part in the manuscript.

11. Fig. 2A. The authors performed growth experiments of selected bacteria and talked about growth rates. Instead, they measured the final Optical Density after 24 hours, when bacteria could have reached similar densities with different rates. This experiment should be better performed by monitoring the kinetics of growth, i.e., OD/time, and then the results interpreted accordingly.

Response: Monitoring the kinetics of bacterial growth (OD/time) would be a better way to observe the effect of PCN on bacteria. Therefore, we repeated the experiments and monitored OD600 of each bacterium and drew a growth curve.

12. Line 168-9. Fig. 2F does not match what is stated in the text.

Response: We have modified the statement about Fig. 2F.

13. How do the authors explain the expansion of *Bifidobacterium longum*, a bacterium that apparently contains *pncA*, upon PCN treatment?

Response: The increase in *Bifidobacterium longum* was unexpected. As shown in Figs. S2 and 2A, *B. longum* had both the *de novo* and salvage pathways, it may prefer previous one, and PncA may not function here. Besides, we think the complex dependence of gut bacteria on each other makes it difficult to judge bacterial fluctuations by changes only in one factor. PCN treatment may have a greater influence on bacteria that interact with *B. longum*.

14. Line 225. "The bacteria were cultured for the indicated period". Indicated where?. At what concentration Nam was used, and at what stage were bacteria collected?

Response: 0.5 mM NAM was added to the culture medium of *E. coli*, and the bacteria were cultured until OD reached 1.0.

15. Line 339. Add "salvage of" before "NA and NAM".

Response: We have added "salvage of" before "NA and NAM" (Line 259).

REVIEWERS' COMMENTS:

Reviewer #1 (Remarks to the Author):

The revision adequately addressed most of my concerns.
The remaining suggestions are detailed below:

1. Fig2 D-F the Y-axes should start from zero. Current presentation is misleading exaggerating the rather small (~10%) differences in NAD levels.

2. The very modest increase of NAD in livers of mice repopulated with *pncA*-OE *E.coli* and the lack of protection in the MCD model might be due to the fact that the authors did not treat the mice with IPTG. Their inducible vector requires IPTG for *pncA* overexpression. The authors should verify the expression of His-*pncA* in feces. The fact that they did not observe increased NA in feces supports the conclusion that their overexpression system is not yet optimized in-vivo. I would suggest repeating these experiments with IPTG treatment or switching to a constitutive bacterial expression vector that does not require an inducer. I understand that these experiments will take a long time and thus I do not consider them as a prerequisite for publication of the current paper but rather as a suggestion for future studies.

Reviewer #2 (Remarks to the Author):

In this revised form, the authors have properly and duly addressed all concerns.

Note: in Fig.1, ftNadM label should be removed as it does not perform the indicated reaction (NaMN->NMN). Only ftNadE does.

Response to Reviewers and Editors

Reviewer 1:

1. Fig2 D-F the Y-axes should start from zero. Current presentation is misleading exaggerating the rather small (~10%) differences in NAD levels.

Response: We have modified the Y-axes of Fig2 D-F.

2. The very modest increase of NAD in livers of mice repopulated with pncA-OE E.coli and the lack of protection in the MCD model might be due to the fact that the authors did not treat the mice with IPTG. Their inducible vector requires IPTG for pncA overexpression. The authors should verify the expression of His-pncA in feces. The fact that they did not observe increased NA in feces supports the conclusion that their overexpression system is not yet optimized in-vivo. I would suggest repeating these experiments with IPTG treatment or switching to a constitutive bacterial expression vector that does not require an inducer. I understand that these experiments will take a long time and thus I do not consider them as a prerequisite for publication of the current paper but rather as a suggestion for future studies.

Response: Thank you for your valuable suggestions. We have discussed this limitation in the end of manuscript and we would check it in our future study.

Reviewer 2:

1. in Fig.1, ftNadM label should be removed as it does not perform the indicated reaction (NaMN->NMN). Only ftNadE does.

Response: We have deleted ftNadM label in Fig. 1.